# Cortico-Hippocampal Computational Modeling Using Quantum Neural Networks to Simulate Classical Conditioning Paradigms

**DOI:** 10.3390/brainsci10070431

**Published:** 2020-07-07

**Authors:** Mustafa Khalid, Jun Wu, Taghreed M. Ali, Thaair Ameen, Ahmed A. Moustafa, Qiuguo Zhu, Rong Xiong

**Affiliations:** 1The State Key Laboratory of Industrial Control Technology, Institute of Cyber-Systems and Control, Zhejiang University, Hangzhou 310027, China; mustafa_khalid@zju.edu.cn (M.K.); qgzhu@zju.edu.cn (Q.Z.); rxiong@iipc.zju.edu.cn (R.X.); 2The Binhai Industrial Technology Research Institute of Zhejiang University, Tianjin 300301, China; 3Electrical Engineering Department, University of Baghdad, Baghdad 10071, Iraq; taghreed.ali@coeng.uobaghdad.edu.iq; 4The Institute of Computer Science, Zhejiang University, Hangzhou 310027, China; thaearm@zju.edu.cn; 5The Marcs Institute for Brain and Behaviour and School of Psychology, Western Sydney University, Sydney 1797, Australia; ahmedhalimo@gmail.com; 6The Department of Human Anatomy and Physiology, the Faculty of Health Sciences, University of Johannesburg, Johannesburg 2198, South Africa

**Keywords:** quantum neural network, computational modeling, classical conditioning, lesioned and intact model, cortico-hippocampal model

## Abstract

Most existing cortico-hippocampal computational models use different artificial neural network topologies. These conventional approaches, which simulate various biological paradigms, can get slow training and inadequate conditioned responses for two reasons: increases in the number of conditioned stimuli and in the complexity of the simulated biological paradigms in different phases. In this paper, a cortico-hippocampal computational quantum (CHCQ) model is proposed for modeling intact and lesioned systems. The CHCQ model is the first computational model that uses the quantum neural networks for simulating the biological paradigms. The model consists of two entangled quantum neural networks: an adaptive single-layer feedforward quantum neural network and an autoencoder quantum neural network. The CHCQ model adaptively updates all the weights of its quantum neural networks using quantum instar, outstar, and Widrow–Hoff learning algorithms. Our model successfully simulated several biological processes and maintained the output-conditioned responses quickly and efficiently. Moreover, the results were consistent with prior biological studies.

## 1. Introduction

The perceptron is the first fundamental model for artificial neural networks (ANNs) proposed by Rosenblatt in 1958. Therefore, the assumption of weighted connections and neurons has been used to simulate biological brain behavior and find optimal solutions for multivariate problems [1].

For decades, ANNs have been considered to be the dominant approach in tasks requiring intelligence such as object classification, natural language programming, data recommendation, and facial recognition. Topologies such as feedforward, recurrent, convolutional, classical, and deep neural networks have been used with various modifications in different applications. The learning process of an ANN is based on the optimization of an assigned performance function using a sequence of iterations to map the output vectors to the related inputs [2,3].

Classical and deep neural networks have been used to simulate human and animal brain regions with distinct functions for decades. Researchers have proposed various models using ANNs to mimic some specific regions of the brain [4]. They validated their proposed models with empirical biological experiments using conditioned stimulus (CS), unconditioned stimulus (US), and conditioned response (CR) [3,5]. Such models use the powerful ability of ANNs to test brain activities with unprecedentedly strong tendencies in humans and animals [6,7].

ANNs, which mimic many real and well-defined systems, have become popular and practical, specifically in biological fields such as computational neuroscience and cognitive modeling. However, an ANN still has some severe deficiencies, such as the inadequacy of its modeling of memory, its dependency on the convergence of the iterative learning, and the diversity of optimization techniques needed to find the optimized network parameters [2,8]. To address these problems, Schrödinger’s quantum equation has been used to generate the quantum neural network (QNN) approach [9].

Consequently, several pioneering quantum computing models have been proposed, such as the quantum computational network of Deutsch [10], the factoring algorithm of Shor [11], and the search algorithm of Grover [12]. Kak introduced the first quantum network that depends on neural network principles [9]. Since the first QNN model was introduced, various other models have been proposed [13].

The possibility of using quantum mechanics for computational modeling was first proposed by Feynman. In 1985, Feynman examined a fundamental quantum model that represents the elementary logical truth tables by changing the spin directions of the electron in terms of quantum mechanics. These type of features inspired researchers to consider a QNN implementation [14].

Brain memory modeling has opened the doors for entangled QNNs due to quantum computing being considered as a powerful tool to accelerate the performance of computational neural network models [15]. Furthermore, the entangled QNN has initiated the concept of using a quantum bit (qubit) instead of the neurons in the neural network [16,17]. Neural networks are based on the idea of the interconnected units, which represent biological neurons. These units feed the input signal to one another using two states: “active” and “inactive” [18]. Rather than use bits in classical computers or neurons in ANN, QNNs use qubits, which are the smallest units that store a circuit’s state during the computation in a quantum network [19] and also have two states of behavior [18].

In this paper, we propose a cortico-hippocampal computational quantum (CHCQ) model that is based on two entangled QNNs to simulate different biological paradigms in intact and lesioned systems. Studying the behavior of both intact and hippocampal-lesioned systems is significant for the multiphase learning paradigms. Most of these paradigms have a similar response at the first phase but respond differently to the other phases. The CHCQ model mainly consists of an adaptive single-layer feedforward QNN (ASLFFQNN), which models the cortical module; and an autoencoder QNN (AQNN), which represents the hippocampal module. The AQNN forwards its internal representations for adaptive learning in the cortical module. We compare the CHCQ model with the Green model [20], which is our recently published model based on classical neural networks. We further compare the results of the CHCQ model with those of the Gluck/Myers model [21] and the model of Moustafa et al. [22] by simulating different biological paradigms, as described in Table 1. We compare our proposed model with the previous models to confirm how the CHCQ model performs a wide range of biological paradigms and classical conditioning tasks effectively.

However, the usage of quantum computation in the CHCQ model makes it more powerful to simulate the cortico-hippocampal region than the classical computation. Instead of simulating the n–bits input cue of information in classical computation, the quantum computation simulates the same cue as $2^{n}$ possible states. The quantum circuit (such as quantum rotation gates) provides computational speedup over the ANNs in classical conditioning simulations. Such accelerated speedup enables the CHCQ model to simulate many biological paradigms efficiently.

This paper proceeds as follows: Initially, we explain the structure of the qubit, quantum rotation gate, and qubit neuron model. Then, the proposed model for both intact and lesioned systems is introduced. Subsequently, we discuss the results of the simulated biological paradigms in Table 1 and compare them with the Gluck/Myers, Moustafa et al., and Green models. Finally, the conclusion summarizes our contributions.

## 2. Materials and Methods

### 2.1. Qubit

The smallest unit that stores information in a quantum computer is called a qubit and is the complement of the classical bit in terms of conventional computers. The two quantum physical states |0〉 and |1〉 denote the classical bit values 0 and 1, respectively. The “|.〉” notation is called Dirac notation and is used to represent the quantum states. In contrast to a bit, a qubit state ϕ forms the superposition of many states as a linear combination, expressed as follows:(1)|ϕ〉=α|0〉+β|1〉,
where α and β are two complex numbers considered to be the probabilities of the |0〉 and |1〉 states, respectively. Rewriting (1) as function form in which the real and imaginary parts are the relative probabilities of |0〉 and |1〉 as follows:(2)$$f\left(\theta \right)=e^{i(\theta )}=\cos\left(\theta \right)+i\sin\left(\theta \right).$$

If there are *n* qubits, then the qubit system (ψ) is a superposition of the following $2^{n}$ or *N* ground states:(3)$$|\psi \rangle =\sum_{n=1}^{N}A_{n}|n\rangle ,$$
where $A_{n}$ is the probability of the related quantum state |n〉. Naturally, the sum of these probabilities equals one, as described in the following equation:(4)$$\sum_{n=1}^{N}{\left|A_{n}\right|}^{2}=1.$$

As a result, |ϕ〉 in (1) collapses into either |0〉 state with probability ${\left|\alpha \right|}^{2}$ or |1〉 state with probability ${\left|\beta \right|}^{2}$. In particular,
(5)$${\left|\alpha \right|}^{2}+{\left|\beta \right|}^{2}=1.$$

### 2.2. Quantum Rotation Gate

Just as there is a logic gate in conventional computation, there is also something similar in quantum computation, which is called a quantum gate. The quantum gate is a qubit state influenced by a series of unitary transforms within a particular interval. In this paper, we use Walsh transform or Hadamard gate (H). This gate produces equal relative probabilities for every qubit to apply superposition process on the qubit system as follows:(6)$$H\left(\alpha |0\rangle +\beta |1\rangle \right)=\alpha \left(\frac{|0\rangle +|1\rangle }{\sqrt{2}}\right)+\beta \left(\frac{|0\rangle -|1\rangle }{\sqrt{2}}\right).$$

### 2.3. Qubit Neuron Model

The proposed qubit neuron model shown in Figure 1 has input–output stages that behave according to the following equation:(7)z=f(y),
(8)$$y=\frac{\pi }{2}S\left(\varepsilon \right)-Arg\left(v\right),$$
(9)$$S\left(\varepsilon \right)=\frac{1}{1+e^{-\varepsilon }},$$
(10)$$v_{j}=\sum_{i}^{R}w_{ij}\cdot f\left(\tilde{p_{i}^{I}}\right)-f\left(\vartheta \right),$$
(11)$$\tilde{p_{i}}=\frac{\pi }{2}p_{i},$$
where function *f* is described in (2); *y* is the output of the network; *S* is the logsigmoid activation function of reversal parameter ε having the range [0, 1]; Arg(v) is the argument function, which returns the phase value of the complex number *v*; $w_{ij}$ is the weight of from the *i*th input ($p_{i}$) to the *j*th hidden qubit, the tilde notation represents the input qubit produced by Hadamard gate; and ϑ is the phase parameter in terms of threshold.

### 2.4. Proposed Model

In Table 1, both A and B are two input CSs, and the associated numbers are their amplitudes. While the suffix X, Y, or even Z are contexts of the related input cue, and the plus (+) and minus (−) signs represents the pairing and unpairing status between CSs and USs. While the prime sign $\left({}^{\prime }\right)$ stands for the related CSs.

The CHCQ model, as shown in Figure 2, principally consists of two QNNs as follows:The hippocampal region has an AQNN that uses both instar and outstar rules to reproduce the inputs and generate the internal representations, which are forwarded to the cortical region in intact systems but not in lesioned ones.The cortical region has an ASLFFQNN that uses both instar and Widrow–Hoff rule to update its weights and quantum parameters.

The instar and outstar rules are learning algorithms developed by [23], and used with normalized input cues; while the Widrow–Hoff rule is a supervised learning algorithm developed by [24] which depends on the desired output.

The CHCQ model was implemented for two systems, as shown in Figure 3. The intact system is the whole proposed CHCQ model, which represents both the cortical and hippocampal networks, whereas the lesioned system is the same model after lesioning the hippocampal module by omitting the connective link that forwards the internal representations from the hidden layer of the AQNN to the ASLFFQNN. The framework is shown in Figure 2, which illustrates the intact system and the lesioned system.

However, the output response of the ASLFFQNN is the measured CR which has a value between 0 and 1. During the learning procedure, the value of the output CR varies depending on its related task and condition. Accordingly, this learning ends when the CR value reaches its desired constant state which is the steady state (either 0 or 1).

#### 2.4.1. General QNN Architecture

The QNNs of both the cortical and hippocampal regions consist of three layers: input, hidden, and output layers.

##### Input Layer (I)

As shown in Figure 4, this layer converts the values of the input cue *p*, which are in the range of [0,1] into the range [0,π/2], which is the range suitable for quantum states using Hadamard gate. Then, it estimates the output according to (7) and (11) as follows:(12)$$z_{i}^{I}=f\left(\tilde{p_{i}^{I}}\right),$$
(13)$$\tilde{p_{i}^{I}}=\frac{\pi }{2}p_{i},$$
where $p_{i}$ is the *i*th input item of the input cue and $\tilde{p_{i}^{I}}$ is the quantum input.

##### Hidden Layer (H)

The hidden layer uses the aforementioned set of (7)–(11) to obtain the output as follows:(14)$$z_{j}^{H}=f\left(y_{j}^{H}\right),$$
(15)$$\begin{array}{cc} v_{j}^{H} & =\sum_{i=1}^{R}w_{ij}\cdot f\left(\tilde{p_{i}^{I}}\right)-f\left(\vartheta_{j}^{H}\right)\\ & =\sum_{i=1}^{R}w_{ij}\cdot e^{i(\tilde{p_{i}^{I}})}-f\left(\vartheta_{j}^{H}\right)\\ & =\sum_{i=1}^{R}w_{ij}\cdot \left(\cos\left(\tilde{p_{i}^{I}}\right)+i\sin\left(\tilde{p_{i}^{I}}\right)\right)-\cos\left(\vartheta_{j}^{H}\right)-i\sin\left(\vartheta_{j}^{H}\right), \end{array}$$
(16)$$\begin{array}{cc} y_{j}^{H} & =\frac{\pi }{2}S\left(\varepsilon_{j}^{H}\right)-Arg\left(v_{j}^{H}\right)\\ & =\frac{\pi }{2}S\left(\varepsilon_{j}^{H}\right)-\mathrm{arctant}\left(\frac{\sum_{i=1}^{R}w_{ij}\cdot \sin\left(\tilde{p_{i}^{I}}\right)-\sin\left(\vartheta_{j}^{H}\right)}{\sum_{i=1}^{R}w_{ij}\cdot \cos\left(\tilde{p_{i}^{I}}\right)-\cos\left(\vartheta_{j}^{H}\right)}\right), \end{array}$$
where $W_{ij}$ is the hidden-layer weight from the *i*th input node to the *j*th hidden qubit for *R* inputs and *Q* qubits.

##### Output Layer (O)

This layer follows a scheme that is similar to that of the previous layer by using the corresponding set of equations to obtain the final output as follows:(17)$$z^{O}=f\left(y^{O}\right),$$
(18)$$\begin{array}{cc} v^{O} & =\sum_{j=1}^{Q}w_{j}\cdot f\left(y_{j}^{H}\right)-f\left(\vartheta^{O}\right)\\ & =\sum_{j=1}^{Q}w_{j}\cdot e^{i(y_{j}^{H})}-f\left(\vartheta^{O}\right)\\ & =\sum_{j=1}^{Q}w_{j}\cdot \left(\cos\left(y_{j}^{H}\right)+i\sin\left(y_{j}^{H}\right)\right)-\cos\left(\vartheta^{O}\right)-i\sin\left(\vartheta^{O}\right), \end{array}$$
(19)$$\begin{array}{cc} y^{O} & =\frac{\pi }{2}S\left(\varepsilon^{O}\right)-Arg\left(v^{O}\right)\\ & =\frac{\pi }{2}S\left(\varepsilon^{O}\right)-\arctan\left(\frac{\sum_{j=1}^{Q}w_{j}\cdot \sin\left(y_{j}^{H}\right)-\sin\left(\vartheta^{O}\right)}{\sum_{j=1}^{Q}w_{j}\cdot \cos\left(y_{j}^{H}\right)-\cos\left(\vartheta^{O}\right)}\right), \end{array}$$
where $W_{j}$ is the hidden-layer weight from the *j*th hidden qubit to the output.

#### 2.4.2. Hippocampal Module Network

The AQNN represents the hippocampal region using fully connected qubits with a single hidden layer network. The AQNN encodes the input data and reproduces it at the output layer to generate the internal representations. The weights and QNN circuit parameters (ε and ϑ) use instar and outstar learning in input and output layers, respectively, to update their values as follows:(20)$$W_{ij}^{instar}\left(k+1\right)=W_{ij}^{instar}\left(k\right)+\mu a_{j}^{h1}\left(k\right)\left(p_{i}^{T}\left(k\right)-W_{ij}^{h1}\left(k\right)\right),$$
(21)$$\varepsilon_{ij}^{instar}\left(k+1\right)=\varepsilon_{ij}^{instar}\left(k\right)+\mu a_{j}^{h1}\left(k\right)\left(p_{i}^{T}\left(k\right)-\varepsilon_{ij}^{h1}\left(k\right)\right),$$
(22)$$\vartheta_{ij}^{instar}\left(k+1\right)=\vartheta_{ij}^{instar}\left(k\right)+\mu a_{j}^{h1}\left(k\right)\left(p_{i}^{T}\left(k\right)-\vartheta_{ij}^{h1}\left(k\right)\right),$$
where μ is a small positive number representing learning rate, $a_{j}^{h1}$ is the hippocampal internal representation output vector with *Q* nodes, *p* is the input cue with *R* qubits, superscript *T* means the transpose of the relevant matrix, and $W_{ij}^{h1}$ is the hippocampal internal-layer weighs from the *i*th input node to the *j*th hidden qubit with *R* qubits and *Q* nodes. Note that (*k*) refers to the current state whereas (*k* + 1) is the succeeding or new state.

However, the internal representations are gathered to reproduce the input cue similarities at the output layer. Then, the AQNN updates the weights and QNN circuit parameters of the output layer, using the outstar learning algorithm through the following set of equations to complete this unsupervised learning process:(23)$$W_{ij}^{outstar}\left(k+1\right)=W_{ij}^{outstar}\left(k\right)+\mu \left(a_{j}^{h2}\left(k\right)-W_{ij}^{h2}\left(k\right)\right)a_{i}^{h1T}\left(k\right),$$
(24)$$\varepsilon_{ij}^{outstar}\left(k+1\right)=\varepsilon_{ij}^{outstar}\left(k\right)+\mu \left(a_{j}^{h2}\left(k\right)-\varepsilon_{ij}^{h2}\left(k\right)\right)a_{i}^{h1T}\left(k\right),$$
(25)$$\vartheta_{ij}^{outstar}\left(k+1\right)=\vartheta_{ij}^{outstar}\left(k\right)+\mu \left(a_{j}^{h2}\left(k\right)-\vartheta_{ij}^{h2}\left(k\right)\right)a_{i}^{h1T}\left(k\right),$$
where $a_{j}^{h2}$ is the actual output vector of the hippocampal side with *R* outputs and $W_{ij}^{h2}$ is the weight from the *i*th hippocampal hidden qubit to the *j*th corresponding output.

#### 2.4.3. Cortical Module Network

This region has a fully connected supervised ASLFFQNN. The input layer is trained by the instar learning rule to adapt the internal mapped representations to the input cue. These representations are generated by the hippocampal side as an adaptive learning signal. First, the hippocampal input layer’s elements (weights and quantum parameters) are initialized randomly for the lesioned system case. Alternatively, the internal elements are initialized by mapping them to their opposing elements on the hippocampal module through an adaptive signal, which carries the adaptive weights and other quantum parameters, as follows:(26)$$W_{ij}^{instar}\left(k+1\right)=W_{ij}^{instar}\left(k\right)+\mu a_{j}^{c1}\left(k\right)\left(p_{i}^{T}\left(k\right)-W_{ij}^{c1}\left(k\right)\right),$$
(27)$$\varepsilon_{ij}^{instar}\left(k+1\right)=\varepsilon_{ij}^{instar}\left(k\right)+\mu a_{j}^{c1}\left(k\right)\left(p_{i}^{T}\left(k\right)-\varepsilon_{ij}^{c1}\left(k\right)\right),$$
(28)$$\vartheta_{ij}^{instar}\left(k+1\right)=\vartheta_{ij}^{instar}\left(k\right)+\mu a_{j}^{c1}\left(k\right)\left(p_{i}^{T}\left(k\right)-\vartheta_{ij}^{c1}\left(k\right)\right),$$
where $a_{j}^{c1}$ is the cortical internal representation output vector with *Q* qubits, *p* is the input cue with *R* inputs, and $W_{ij}^{c1}$ is cortical internal layer weight from the *i*th input node to the *j*th hidden qubit with *R* inputs and *Q* qubits.

Finally, the cortical upper layer uses the Widrow–Hoff learning algorithm to update the weights and quantum parameters at the output layer as follows:(29)$$W_{ij}^{c2}\left(k+1\right)=W_{ij}^{c2}\left(k\right)+\mu a_{i}^{c1}\left(k\right)e_{j}\left(k\right),$$
(30)$$\varepsilon_{ij}^{c2}\left(k+1\right)=\varepsilon_{ij}^{c2}\left(k\right)+\mu a_{i}^{c1}\left(k\right)e_{j}\left(k\right),$$
(31)$$\vartheta_{ij}^{c2}\left(k+1\right)=\vartheta_{j}^{c2}\left(k\right)+\mu a_{i}^{c1}\left(k\right)e_{j}\left(k\right),$$
(32)$$MAE=\frac{1}{2}\sum_{j=1}^{iter}e_{j}=\frac{1}{2}\sum_{j=1}^{iter}\left(y_{j}-d_{j}\right),$$
where $W_{i,j}^{c2}$ is the top-layer weight matrix of the cortical side from input $p_{i}$ to output nodes $y_{j}$, $d_{j}$ is the desired output or US, and MAE is the mean absolute error between the actual and desired *j*th output, respectively.

## 3. Results

The CHCQ model was tested using the tasks listed in Table 1 for both intact and lesioned systems. The results are compared with those of the Moustafa et al., Gluck/Myers, and Green models. For each task, we measured the CR value after reaching the steady state in each trial to obtain the final response. The final stable computed CR values should be either 1 or 0 for CS+ or CS− learning, respectively.

### 3.1. Primitive Tasks

Basically, the CHCQ model was trained to pair and impair only one CS with a US in addition to the context. For both of the simple tasks (AX+ and AX−), only one learning phase was used to obtain the output final response. The CR of the A+ stimuli was rapidly obtained and directly reached the steady state within a shorter time than it did when the model of Moustafa et al. was used, as shown in Figure 5 and confirmed by the experiments of [25,26,27]. Likewise, Figure 6 shows how fast the CR of the A− stimuli and the related context obtained the zero final state. The CHCQ model took a few numbers of trials to reach the steady state for A+ and took fewer trials for A− than did the other models, as shown in the results in Table 2, Table 3 and Table 4.

Similarly, Figure 7 compares the CR of the A+ stimuli for both intact and lesioned systems obtained by the CHCQ and the model of Moustafa et al. Clearly, our model can complete learning efficiently, directly, and quickly. In addition, the CR of the lesioned system was still more quickly obtained than that of the intact system, as confirmed by the experiments of [28,29,30,31,32].

### 3.2. Stimulus Discrimination

Similarly to the previous task, the stimulus discrimination task consisted of one phase to discriminate between two different stimuli, but it synchronized the CSs according to their responses to the US. The following CSs, A+ and B− (or <A+, B−>), were simulated for both intact and lesioned systems. Subsequently, the CHCQ model was capable of discriminating the two different CSs due to their CRs, which reached the final states quickly and efficiently, as shown in Figure 8.

We note that the CHCQ model not only discriminated the two CSs successfully, but also obtained a quick response in the lesioned system. Moreover, the CRs of both A+ and B− reached their final states by fewer trial numbers than their opposing elements in the intact system. However, learning was accelerated in the lesioned system, which made it more responsive than the intact system, as confirmed by the experiments of [33,34,35]. The output CR of the CHCQ model was simulated with noisy and disruptive internal representations to test the stimulus discrimination ability under such conditions. Although the CHCQ model required more trials than for regular discrimination, it discriminated the two different CSs successfully and obtained the desired final state efficiently, as shown in Figure 9.

### 3.3. Discrimination Reversal

Starting from the concept of the previous task, it is worth studying the reverse effects of the training set on the output CR after discriminating them successfully. Thus, the reverse discrimination task was simulated by first training for stimulus discrimination (<A+, B−>) before beginning the reversal task <A−, B+>.

The intact system was initially trained with an <A+, B−> training set, then with <A−, B+>. The results show that discrimination between A and B was completed by fewer trial numbers than the stimulus discrimination task. This indicates that after training the network to differentiate A and B and then doing the reverse, there is an exchange in the CS status and additional learning, which led to getting fewer trial numbers than that in the CS discrimination task, as shown in Figure 10.

By performing the same task in the lesioned system, we revealed that the lesioned system learned quicker than the intact system. In contrast to the intact system, the second phase showed a longer response to discriminate the two CSs, as evident in Figure 10. However, the CHCQ model’s results of this task meet the empirical conclusions of [36,37].

### 3.4. Blocking

The blocking task consists of three consequent phases: <A+>, <AB+>, and <B−>. The first one exists specifically to initiate the model with a pre-exposure CS. The second phase is a compound of different CSs to block one of them, whereas the last phase is for the blocked CS.

Figure 11 shows the CR of the final phase of learning in the intact system; which presented the blocking effect due to the prior conditioning in the preceding phases, as confirmed experimentally by [38,39,40]. The blocking effect can be eliminated by extending the conditioning of the second phase, which made it longer, as shown in Figure 12 and confirmed experimentally by [38,41,42]. Although lesioning the CHCQ model did not affect the output CR, as shown in Figure 13, this is supported by the experimental findings of [43,44,45].

### 3.5. Overshadowing

The overshadowing task consists of only two phases. The first phase is the compound learning of two or more CSs; one of them is more efficient than the other, which produces a remarkable CR despite the rest of the CSs.

However, the model was trained by <AB+>first, then by <A+>and <B+>together. As a result, the intact system performed this task efficiently by simulating A with fewer trial numbers than B after pairing them with the US successfully, as shown in Figure 14 and confirmed experimentally by [46,47,48].

Similar to its performance in the blocking task, the lesioned system did not affect by the overshadowing, as shown in Figure 15 and confirmed by [43]. Expanding the pre-exposure phase could also eliminate the overshadowing effect, as shown in Figure 16, which is supported by the experimental findings of [49,50].

### 3.6. Easy–Hard Transfer

This task has the same concept of the simple or stimulus discrimination tasks, but is for an input cue with different amplitudes within the range of [0, 1]. Such cues could be heating levels, a spectrum of frequencies, or any related inputs that might have gradient values.

Supposing our model needs to recognize two similar CSs, a stimulus with an amplitude greater than 0.5 is considered as a CS–US paired condition (e.g., <A+>). In contrast, those with a value less than 0.5 are considered to be CS–US unpaired conditions for the same stimulus (e.g., <A−>).

We started with an easy task by assigning a stimulus value of 0.9 when a paired condition existed and 0.1 when it did not. Figure 17 and Figure 18 show the output CRs of both intact and lesioned systems for the easy transfer task. Likewise, we simulated a harder transfer task by narrowing the difference between the two levels, which were changed to 0.6 and 0.4, respectively.

After training the CHCQ model with a training set of <A = 0.9+, A = 0.1−> (easy transfer learning), the expected output CRs of the intact and lesioned systems in Figure 17 and Figure 18 were approximately similar to the outputs of the stimulus discrimination task shown in Figure 8.

Both intact and lesioned systems successfully completed the easy transfer stimulus discrimination within a short time or a few trials. In addition, they successfully finished the hard transfer task with a more difficult training set of <A = 0.6+, A = 0.4−>, as shown in Figure 19 and Figure 20, which are approximately similar to the outputs in Figure 8. Moreover, for this task, Figure 21 compares the speed and efficiency of the control and experimental results of the CHCQ model with those of the Green model. Generally, the CHCQ model’s results for this task are confirmed by the experimental studies of [51,52,53,54,55,56,57].

### 3.7. Latent Inhibition

The latent inhibition task consists of two learning phases, which are an unreinforced pre-exposure (<A−>) followed by CS–US pairing phase (<A+>). Figure 22 shows the output CRs of the intact system for both A+ learning with and without pre-exposure to <A−>. Lesioning the model did not affect the output CR of the second phase after the pre-exposure at the beginning, as shown in Figure 23. All of these results were confirmed experimentally by [58,59,60].

Figure 24 and Figure 25 show the CRs of the second phase for both intact and lesioned systems, respectively, and compares them with the results obtained by the Green model. Clearly, the CR response of the CHCQ model took fewer trial numbers and reached the final state directly. Moreover, it was confirmed that the lesioned CR results required fewer trial numbers than those in the intact system.

### 3.8. Generic Feedforward Multilayer Network

As mentioned in the latent inhibition task, lesioning the model did not affect the output CR. To prove that, the CHCQ output of the lesioned system has been compared with the generic multilayer feedforward supervised network [61].

As shown in Figure 26, the CHCQ model successfully obtained the desired output. Additionally, the CHCQ output showed a similar response to the output of the generic feedforward network regardless of the pretraining phase at the beginning.

### 3.9. Sensory Preconditioning

We simulated this task by combining two CSs, let us say A with B, and pairing them in the same training phase <AB−>. Later in the second phase, a conditioned process was applied to predict the US by A only as the <A+> training set. Finally, the second phase conditioning was strengthened by the <B−> training phase to distinguish the effect of the <A+> training set.

Figure 27 shows the output CRs of all three phases, as confirmed by [62,63,64,65,66,67,68,69]. The effect of preconditioning on the CR value of the last trial in the third phase is clear. Figure 28 specifically shows the CR of the last phase. However, lesioning the CHCQ model had no effect on the sensory preconditioning task, as shown in Figure 29 and confirmed by [45,70].

### 3.10. Compound Preconditioning

The compound preconditioning task was simulated in the intact system by compounding the second and third phases of the sensory preconditioning task and integrating them together in the second phase. It became more difficult for the model to discriminate A and B in this task. Figure 30 shows how much longer the compound discrimination needed without the preconditioning phase. Although the task took relatively longer to complete, the proposed model took fewer trial numbers than that of Moustafa et al., as shown in Figure 31. In addition, as in the previous task, lesioning the CHCQ model had no effect on the compound preconditioning task. Generally, the results of the compound preconditioning task have been confirmed by the experimental research of [68,69,71].

### 3.11. Context Sensitivity

The context of the input cue was changed after a number of training trials by shifting its value randomly to obtain a different context in the second phase. As a result, the context shifting task slowed the learning in the intact system without affecting the speed of the lesioned one, as shown in Figure 32. Subsequently, the output CR values shown in Figure 33 prove that the CHCQ model behaved more efficiently. Figure 34 shows the effect of expanding the first phase training number, which abolished the context sensitivity effect. All of these results have been confirmed by the experimental research of [72,73,74,75].

In addition, the CR response was affected, simulating the context sensitivity of the latent inhibition in the intact system. The context sensitivity reduces the learning behavior of the acquisition phase that increases the predicted CR value in the first trial of the second phase, as shown in Figure 35.

## 4. Conclusions

In this paper, we proposed the adaptive CHCQ model shown in Figure 2 and Figure 3. The CHCQ model is the first computational model which uses quantum computation techniques for simulating biological paradigms of classical conditioning. The CHCQ model consists of two main parts: a hippocampal region, represented as an AQNN; and a cortical region, represented as an ASLFFQNN. The AQNN uses quantum instar and outstar learning algorithms to update the weights and QNN circuit parameters to generate the internal representations that are adaptively forwarded to the internal layer of the ASLFFQNN. The ASLFFQNN uses the instar learning algorithm in the input layer and the Widrow–Hoff learning algorithm in the upper layer to update the weights and QNN circuit parameters. The CHCQ model was shown to simulate all the biological paradigms listed in Table 1 successfully with a very efficient and rapid output CR. The quantum circuit provides computational speedup over the ANNs in classical conditioning simulations. The convincing parallel computing features provide an advantage to QNN over ANN.

However, the results presented notable enhancements approved by experimental studies for various tasks that outperform the previously published models. A comparison of the CHCQ model with **M1** (Gluck/Myers model) [21], **M2** (Mustafa et al. model) [22], and **G** (Green model) [20] showed that the CHCQ model has a fast and reliable output CR. Table 2 shows that all the CRs of the CHCQ model reached the final desired states directly after fewer trials than were needed by Gluck/Myers model. Similarly, Table 3 proves that the CHCQ model takes even fewer trial numbers than that of Moustafa et al. Moreover, the CHCQ model completed all of its tasks with more reliable results and more quickly than the Green model, as shown in Table 4. It is worth mentioning that the multiphase learning paradigms of the CHCQ model have no improvement in the first phase as compared to the other models as long as the first phase of such tasks acts as a pre-exposure phase, which takes a similar period in other models.

Our future work will be about building a quantum controller for getting more stable output CRs using delay matching to sample task theory.

## Figures and Tables

**Figure 1 brainsci-10-00431-f001:**
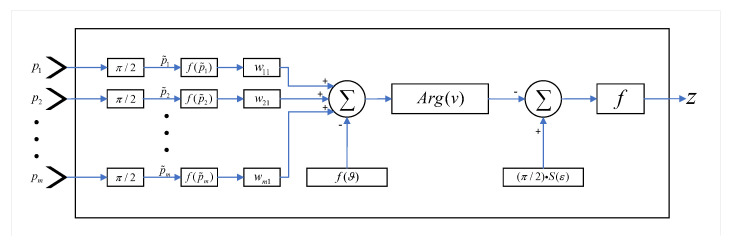
The general structure of the qubit neuron model.

**Figure 2 brainsci-10-00431-f002:**
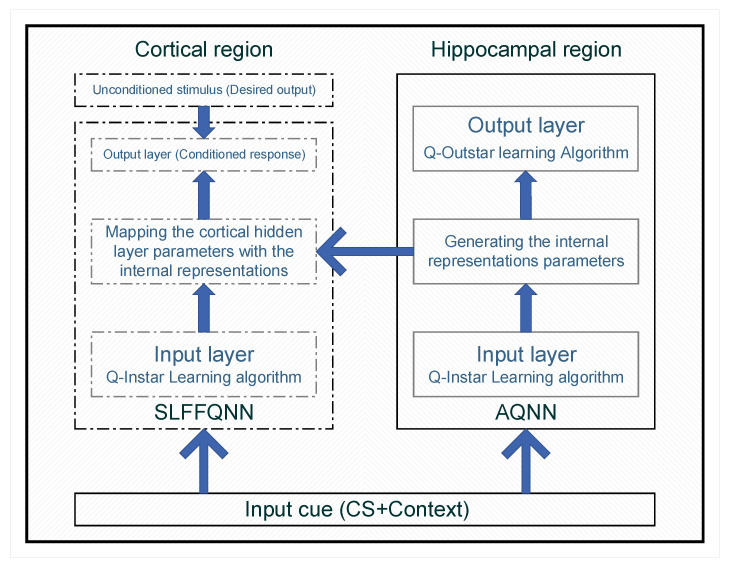
The CHCQ model framework.

**Figure 3 brainsci-10-00431-f003:**
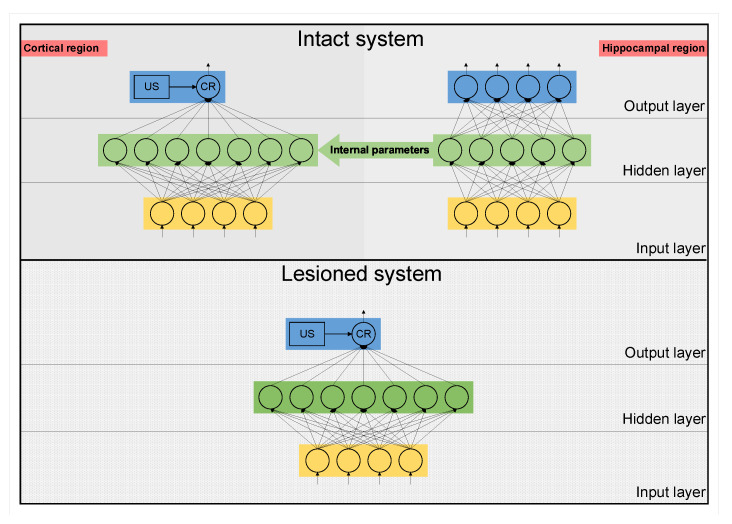
Intact and lesioned systems of the CHCQ model.

**Figure 4 brainsci-10-00431-f004:**
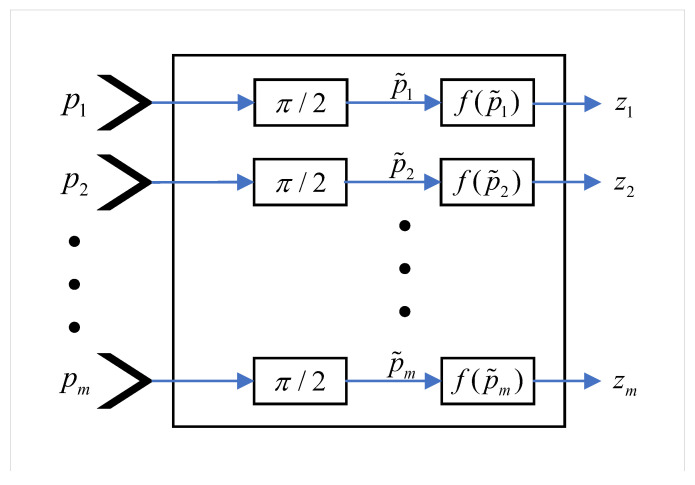
The qubit neuron model in the input layer of CHCQ model.

**Figure 5 brainsci-10-00431-f005:**
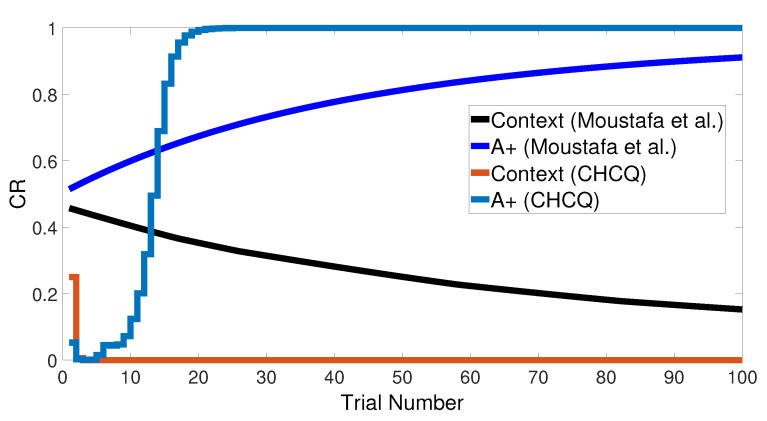
The cue contains context and stimulus A as <A+> task.

**Figure 6 brainsci-10-00431-f006:**
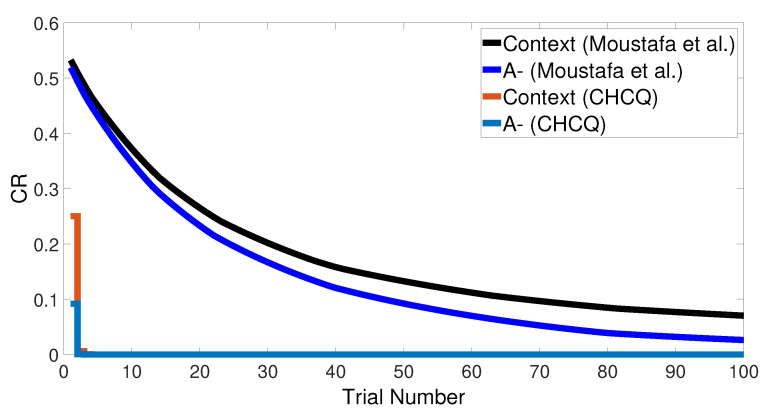
The cue contains context and stimulus A only as <A−> task.

**Figure 7 brainsci-10-00431-f007:**
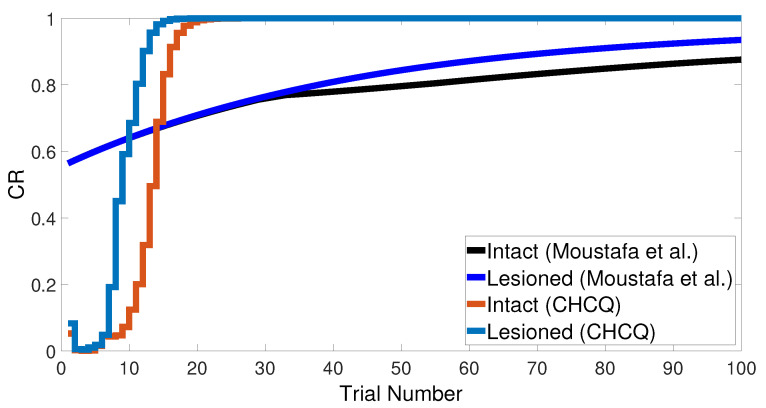
<A+> task for both intact and lesioned systems.

**Figure 8 brainsci-10-00431-f008:**
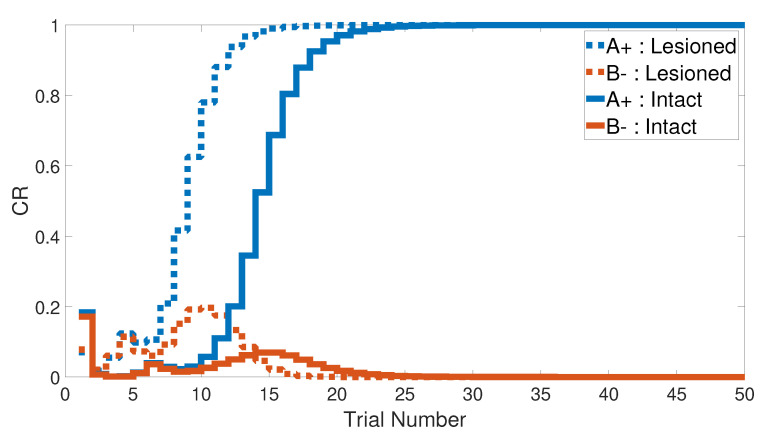
Stimulus discrimination learning <A+, B−> for both intact and lesioned systems of CHCQ model.

**Figure 9 brainsci-10-00431-f009:**
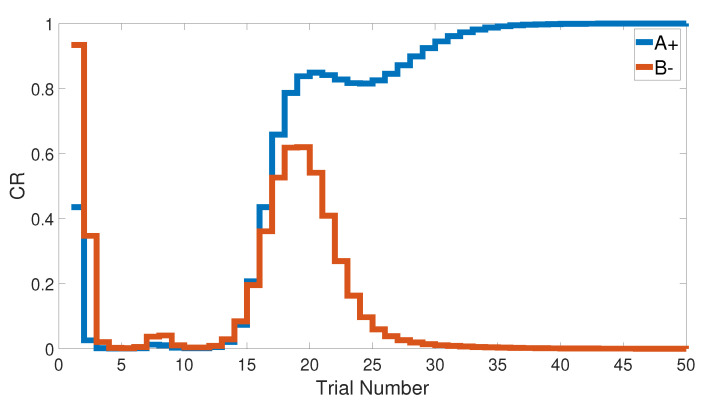
Stimulus discrimination task of the CHCQ-disrupted system.

**Figure 10 brainsci-10-00431-f010:**
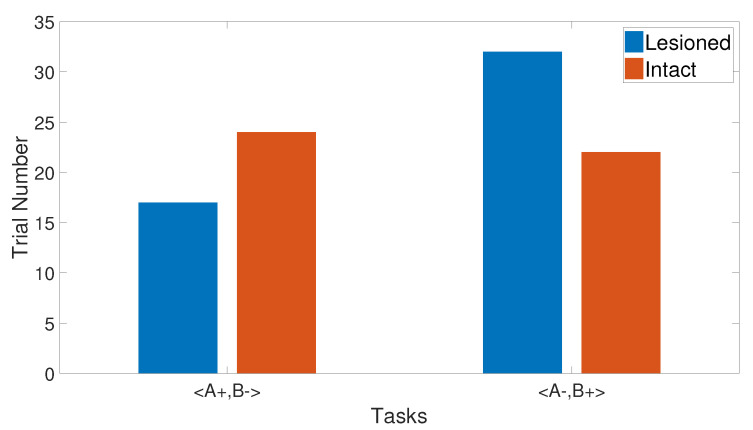
Discrimination reversal for both intact and lesioned systems of CHCQ model.

**Figure 11 brainsci-10-00431-f011:**
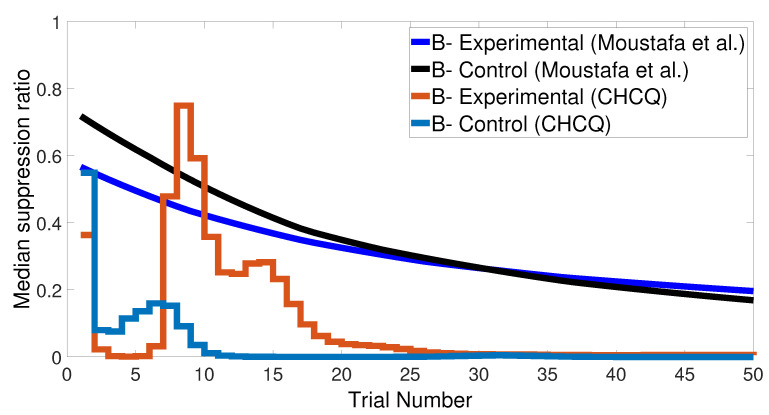
Blocking task for the intact system.

**Figure 12 brainsci-10-00431-f012:**
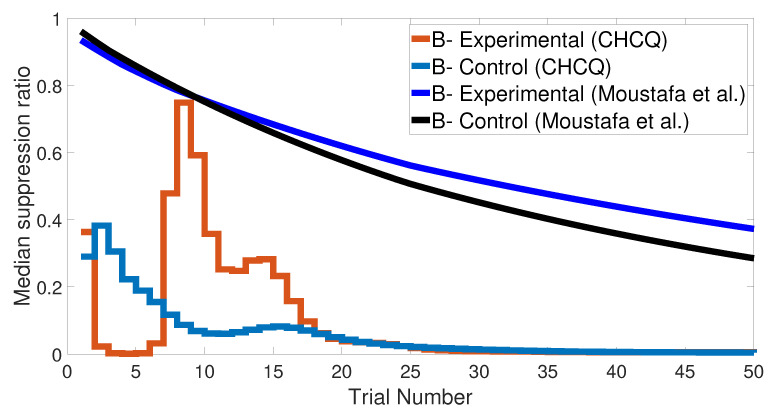
Extended blocking task for the intact system.

**Figure 13 brainsci-10-00431-f013:**
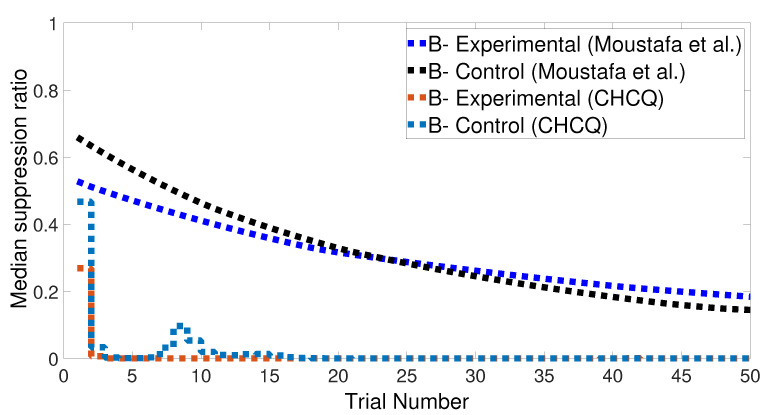
Blocking task for the lesioned system.

**Figure 14 brainsci-10-00431-f014:**
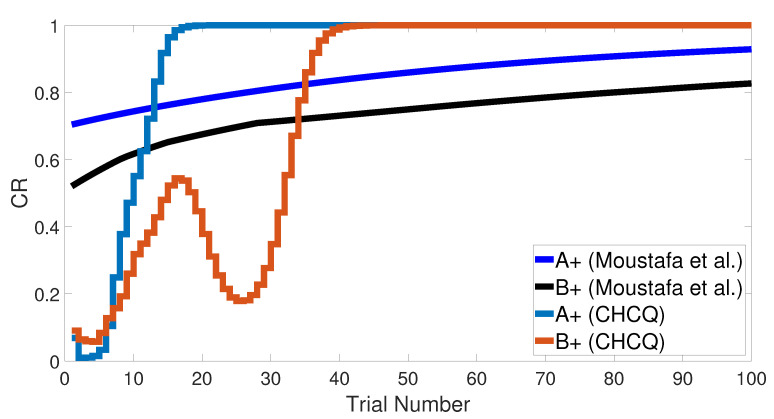
Overshadowing task for the intact system.

**Figure 15 brainsci-10-00431-f015:**
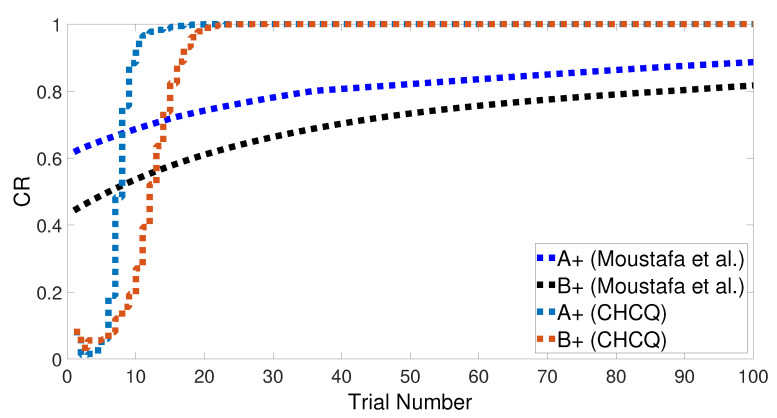
Overshadowing task for the lesioned system.

**Figure 16 brainsci-10-00431-f016:**
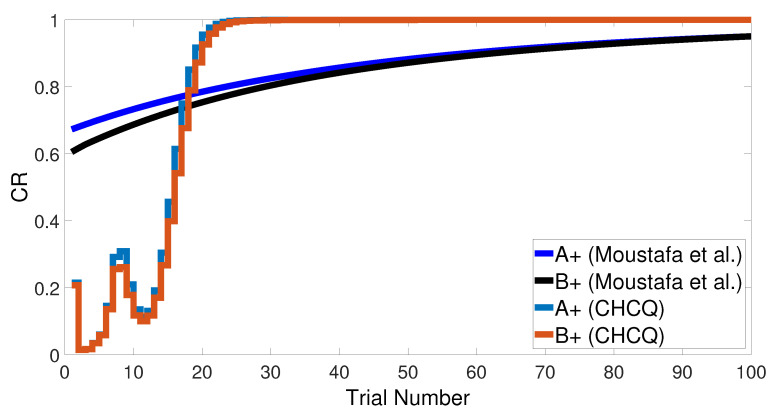
Extended overshadowing task for the intact system.

**Figure 17 brainsci-10-00431-f017:**
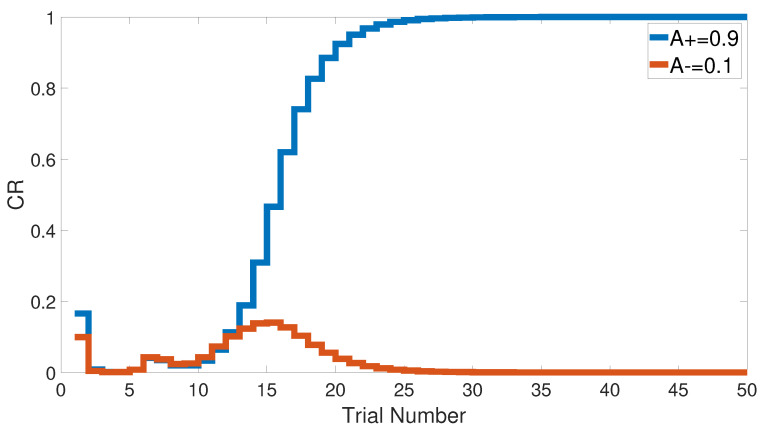
Easy transfer task for the intact system.

**Figure 18 brainsci-10-00431-f018:**
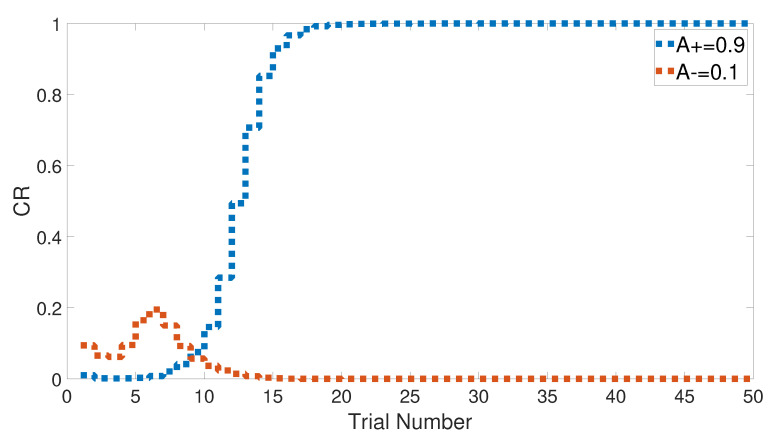
Easy transfer task for the lesioned system.

**Figure 19 brainsci-10-00431-f019:**
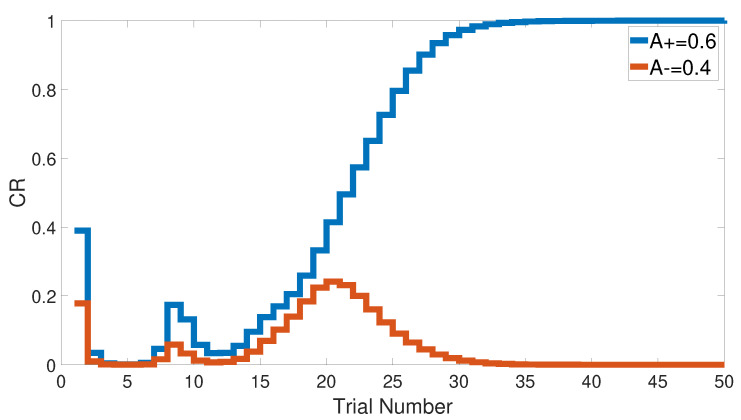
Hard transfer task for the intact system.

**Figure 20 brainsci-10-00431-f020:**
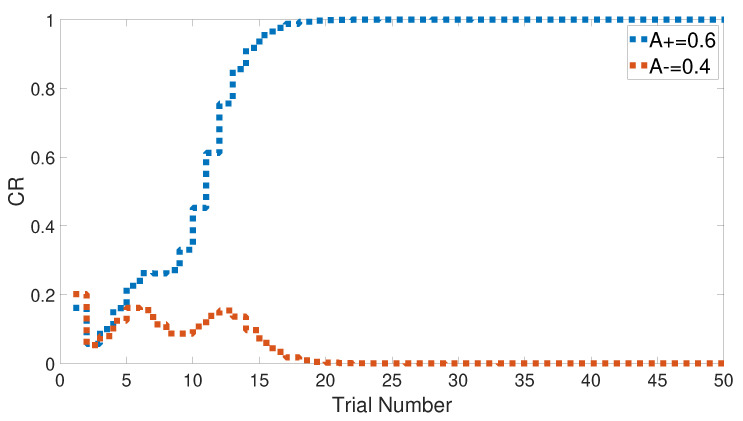
Hard transfer task for the lesioned system.

**Figure 21 brainsci-10-00431-f021:**
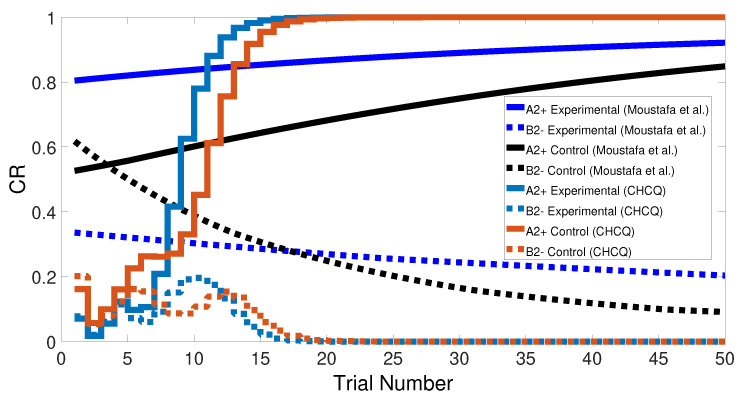
Easy–Hard transfer task of CHCQ model compared with the model of Moustafa et al.

**Figure 22 brainsci-10-00431-f022:**
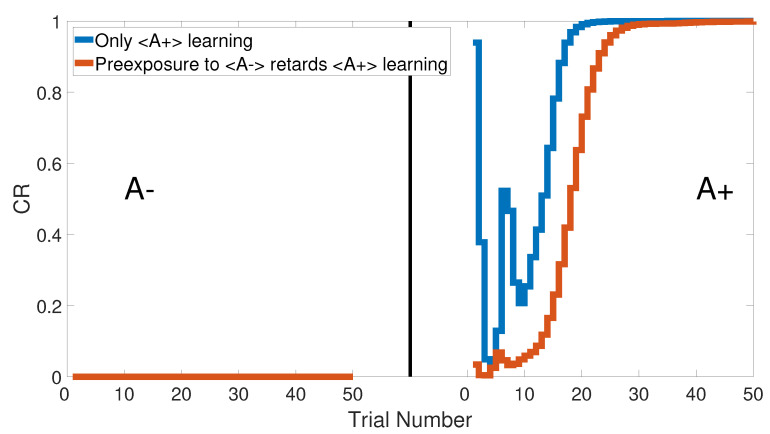
Latent inhibition task for the intact system of CHCQ model with and without a pre-exposure.

**Figure 23 brainsci-10-00431-f023:**
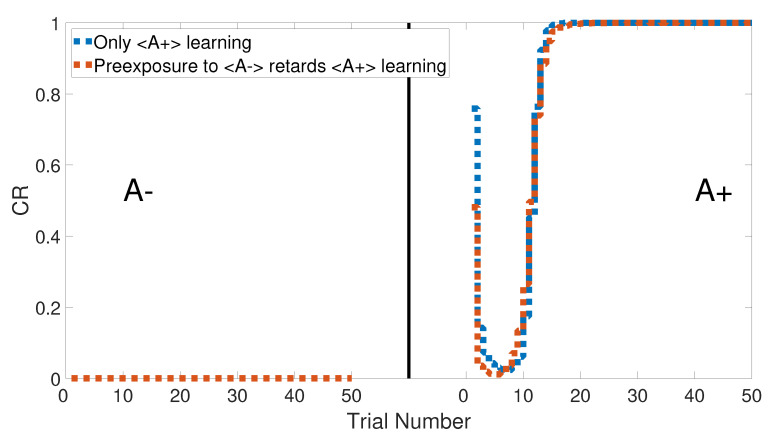
Latent inhibition task for the lesioned system of CHCQ model with and without a pre-exposure.

**Figure 24 brainsci-10-00431-f024:**
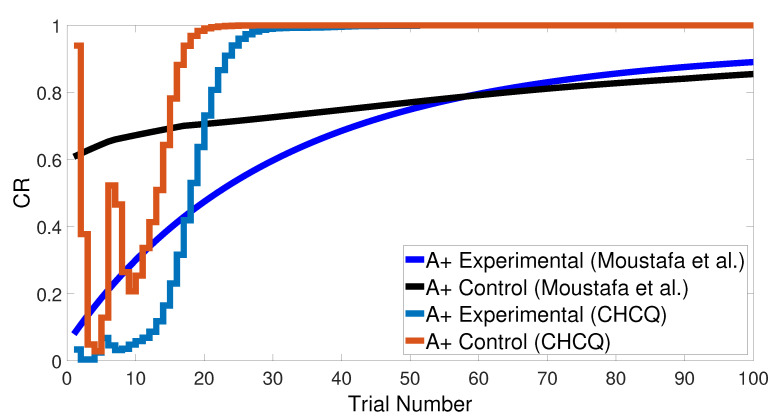
Latent inhibition task for the intact system.

**Figure 25 brainsci-10-00431-f025:**
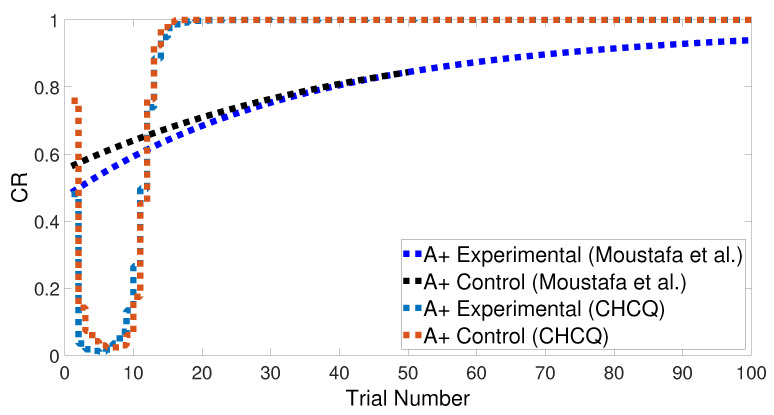
Latent inhibition task for the lesioned system.

**Figure 26 brainsci-10-00431-f026:**
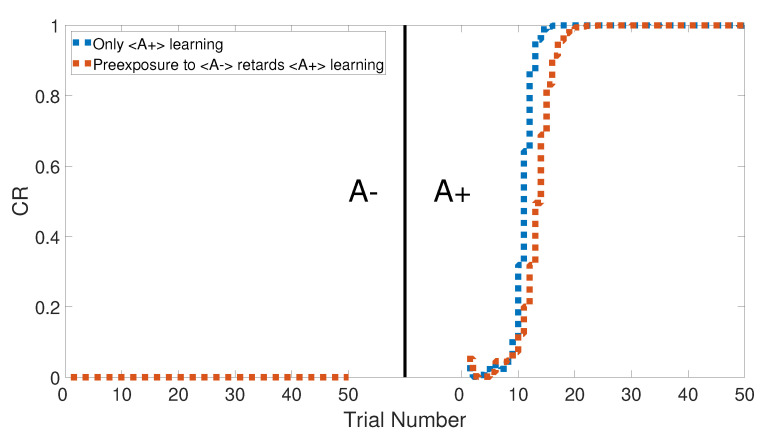
Generic feedforward multilayer network compared to the CHCQ lesioned system.

**Figure 27 brainsci-10-00431-f027:**
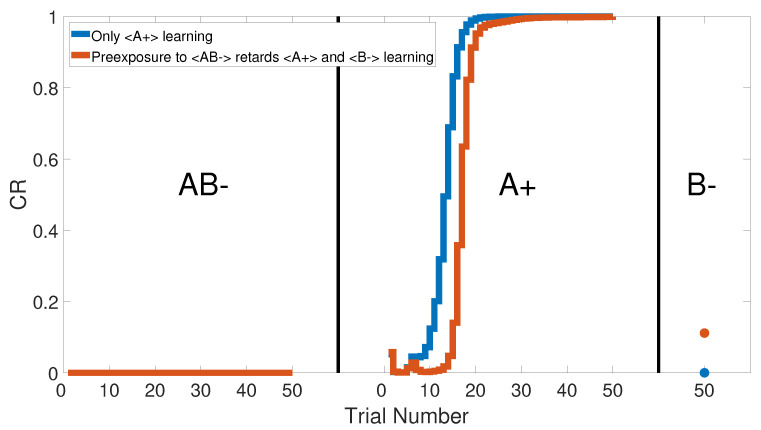
All three phases of sensory preconditioning task for the intact system of the CHCQ.

**Figure 28 brainsci-10-00431-f028:**
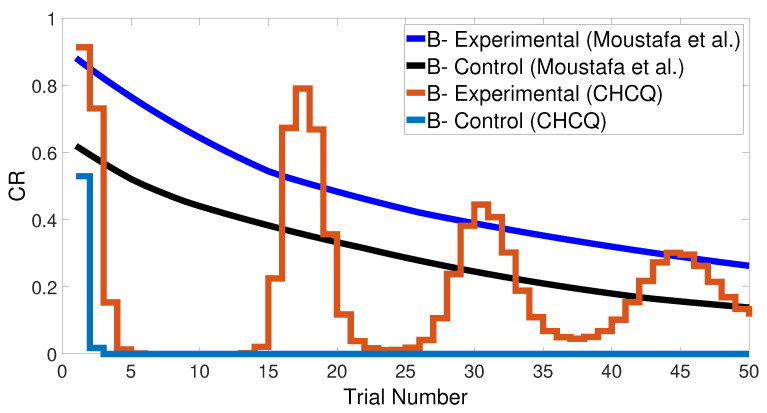
The response of the last phase of sensory preconditioning task for the intact system.

**Figure 29 brainsci-10-00431-f029:**
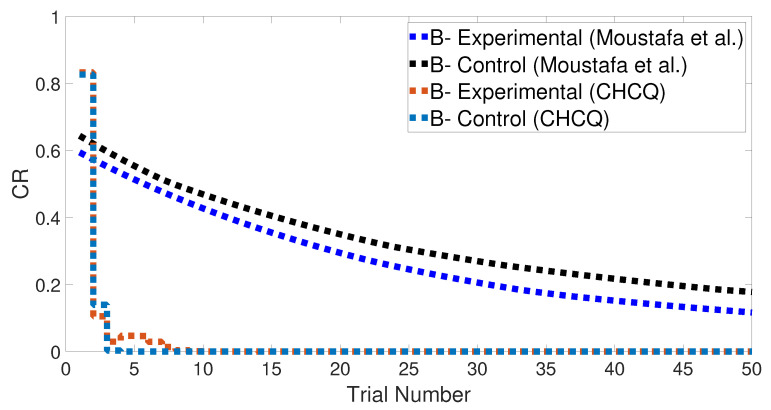
The response of the last phase of sensory preconditioning task for the lesioned system.

**Figure 30 brainsci-10-00431-f030:**
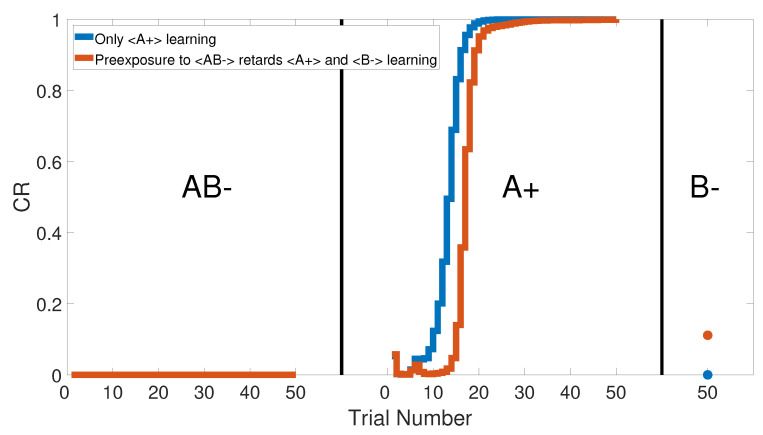
Two phases of compound preconditioning task for the intact system of CHCQ.

**Figure 31 brainsci-10-00431-f031:**
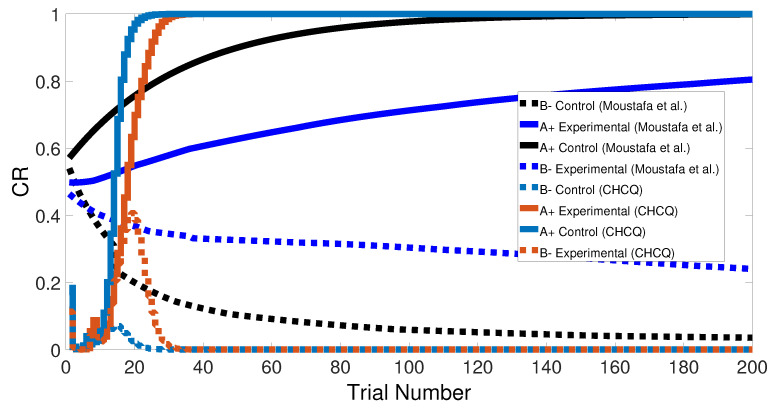
Only the last phase of compound preconditioning task for the intact system.

**Figure 32 brainsci-10-00431-f032:**
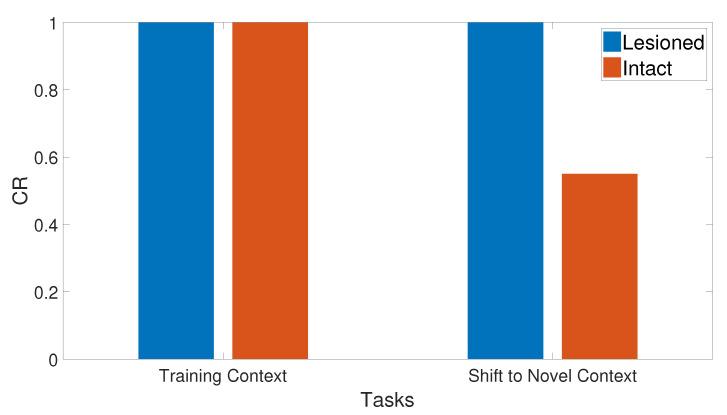
Context sensitivity task for both intact and lesioned systems of the CHCQ model.

**Figure 33 brainsci-10-00431-f033:**
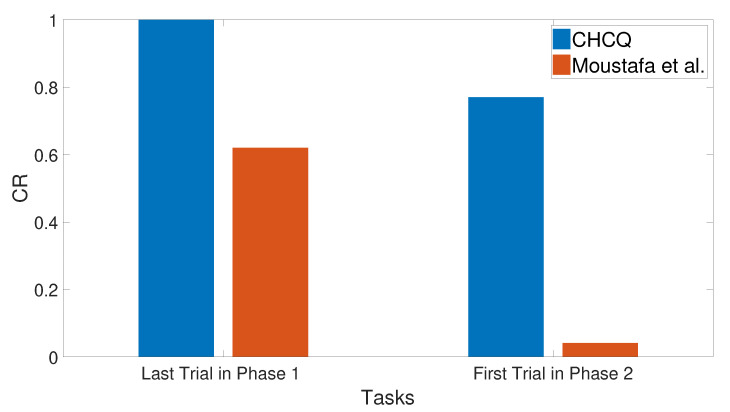
Context sensitivity task for the intact system of the last and first trial values of the two phases, respectively.

**Figure 34 brainsci-10-00431-f034:**
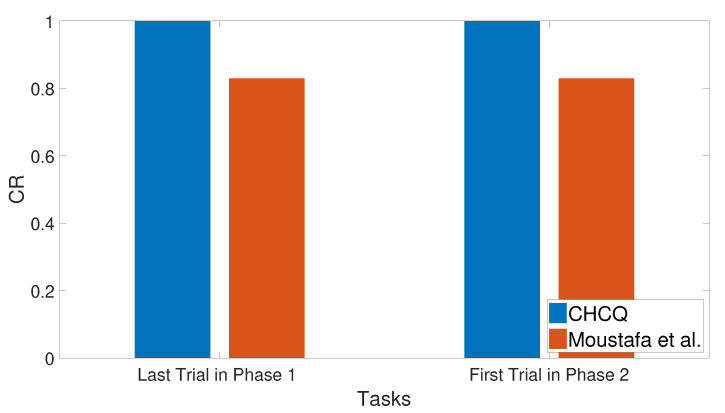
Extended context sensitivity task for the intact system.

**Figure 35 brainsci-10-00431-f035:**
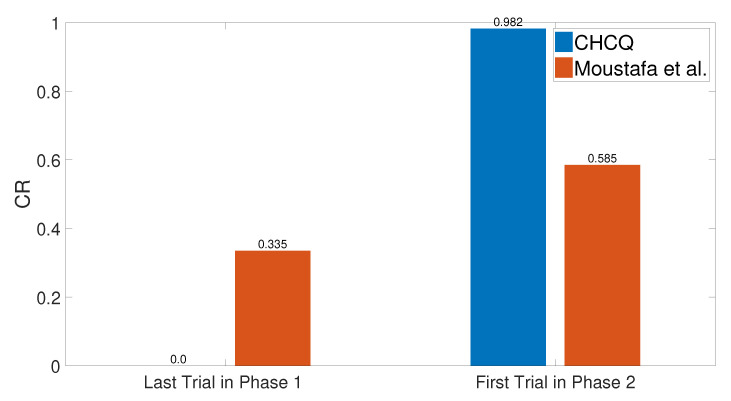
Context shift due to latent inhibition task for the intact system.

**Table 1 brainsci-10-00431-t001:** All the simulated tasks.

No.	Task Name	Phase 1	Phase 2	Phase 3
1A	A+	AX+	—	—
1B	A−	AX−	—	—
2	Stimulus discrimination	AX+, BX−	—	—
3	Discrimination reversal	AX+, BX−	AX−, BX+	—
4	Blocking	AX+	ABX+	BX−
5	Overshadowing	ABX+	AX+; BX+	—
6	Easy–Hard transfer	A1X+, A2X−	A3X+, A4X−	—
7	Latent inhibition	AX−	AX+	—
8	Sensory preconditioning	ABX−	AX+	BX−
9	Compound preconditioning	ABX−	AX+, BX−	—
10A	Context sensitivity (Context shift)	AX+	AY+	—
10B	Context sensitivity of latent inhibition	AX−	AY+	—
11	Generic feedforward multilayer network	AX−	AX+	—

**Table 2 brainsci-10-00431-t002:** A comparison between **M1** (Gluck/Myers model) and the **CHCQ** model, in terms of trial numbers needed to get the final state of conditioned response (CR).

		Phase 1						Phase 2						Phase 3					
		Intact		Lesioned				Intact		Lesioned				Intact		Lesioned			
		(a)	(b)	(c)	(d)	Improvement		(e)	(f)	(g)	(h)	Improvement		(i)	(j)	(k)	(l)	Improvement	
No.	Task Name	M1	CHCQ	M1	CHCQ	b vs. a	d vs. c	M1	CHCQ	M1	CHCQ	f vs. e	h vs. g	M1	CHCQ	M1	CHCQ	j vs. i	l vs. k
1	Stimulus discrimination	>200	24	>200	17	88.0%	91.5%	—	—	—		—	—	—	—	—	—	—	—
2	Reversal learning	>200	24	>200	17	88.0%	91.5%	>200	22	>400	32	89.0%	92.0%	—	—	—	—	—	—
3	Easy–Hard transfer learning	>200	27	>200	19	86.5%	90.5%	>1000	34	>1000	20	96.6%	98%	—	—	—	—	—	—
4	Latent inhibition	50	50	50	50	00.0%	00.0%	>100	31	>100	18	69.0%	82.0%	—	—	—	—	—	—
5	Sensory preconditioning	200	50	—	—	75.0%	—	>100	31	—	—	69.0%	—	50	50	—	—	00.0%	—
6	Compound preconditioning	20	20	—	—	00.0%	—	>100	32	—	—	68.0%	—	—	—	—	—	—	—
7	Generic feedforward multilayer network	—	—	100	50	—	50.0%	—	—	>200	21	—	89.5%	—	—	—	—	—	—
8	Contextual sensitivity	>200	24	>200	17	88.0%	91.5%	>200	1	>200	1	99.5%	99.5%	—	—	—	—	—	—

**Table 3 brainsci-10-00431-t003:** A comparison between **M2** (Moustafa et al. model) and the **CHCQ** model, in terms of trial numbers needed to get the final state of CR.

		Phase 1						Phase 2						Phase 3					
		Intact		Lesioned				Intact		Lesioned				Intact		Lesioned			
		(a)	(b)	(c)	(d)	Improvement		(e)	(f)	(g)	(h)	Improvement		(i)	(j)	(k)	(l)	Improvement	
No.	Task Name	M2	CHCQ	M2	CHCQ	b vs. a	d vs. c	M2	CHCQ	M2	CHCQ	f vs. e	h vs. g	M2	CHCQ	M2	CHCQ	j vs. i	l vs. k
1	A+	>100	23	>100	18	77.0%	82.0%	—	—	—	—	—	—	—	—	—	—	—	—
2	A−	>100	2	>100	2	98.0%	98.0%	—	—	—	—	—	—	—	—	—	—	—	—
3	Sensory preconditioning	100	50	—	—	50.0%	—	>100	31	—	—	69.0%	—	50	50	—	—	00.0%	—
4	Latent inhibition	50	50	50	50	00.0%	00.0%	>100	31	>100	18	69.0%	82.0%	—	—	—	—	—	—
5	Context shift	100	50	—	—	50.0%	—	1	1	—	—	00.0%	—	—	—	—	—	—	—
6	Context sensitivity of latent inhibition	100	50	—	—	50.0%	—	1	1	—	—	00.0%	—	—	—	—	—	—	—
7	Easy–Hard transfer learning	>100	17	—	—	83.0%	—	>100	19	—	—	81.0%	—	—	—	—	—	—	—
8	Blocking	>100	23	>100	18	77.0%	82.0%	>100	24	>100	17	76.0%	83.0%	>100	12	>100	3	88.0%	97.0%
9	Compound preconditioning	100	20	—	—	80.0%	—	>200	32	—	—	84.0%	—	—	—	—	—	—	—
10	Overshadowing	100	20	100	20	80.0%	80.0%	>100	25	>100	22	75.0%	78.0%	—	—	—	—	—	—

**Table 4 brainsci-10-00431-t004:** A comparison between **G** (Green model) and the **CHCQ** model, in terms of trial numbers needed to get the final state of CR.

		Phase 1						Phase 2						Phase 3					
		Intact		Lesioned				Intact		Lesioned				Intact		Lesioned			
		(a)	(b)	(c)	(d)	Improvement		(e)	(f)	(g)	(h)	Improvement		(i)	(j)	(k)	(l)	Improvement	
No.	Task Name	G	CHCQ	G	CHCQ	b vs. a	d vs. c	G	CHCQ	G	CHCQ	f vs. e	h vs. g	G	CHCQ	G	CHCQ	j vs. i	l vs. k
1A	A+	32	23	28	18	28.1%	35.7%	—	—	—	—	—	—	—	—	—	—	—	—
1B	A−	2	2	2	2	00.0%	00.0%	—	—	—	—	—	—	—	—	—	—	—	—
2	Stimulus discrimination	33	24	20	17	27.2%	15.0%	—	—	—	—	—	—	—	—	—	—	—	—
3	Discrimination reversal	33	24	20	17	27.2%	15.0%	31	22	38	32	29.0%	15.7%	—	—	—	—	—	—
4	Blocking	32	23	28	18	28.1%	35.7%	32	24	28	17	25.0%	39.2%	23	12	7	3	47.8%	57.1%
5	Overshadowing	20	20	20	20	00.0%	00.0%	26	25	27	22	03.8%	18.5%	—	—	—	—	—	—
6	Easy–Hard transfer	35	27	25	19	82.5%	87.5%	38	34	27	20	10.5%	25.9%	—	—	—	—	—	—
7	Latent inhibition	50	50	50	50	00.0%	00.0%	41	31	24	18	24.3%	25.0%	—	—	—	—	—	—
8	Sensory preconditioning	50	50	—	—	00.0%	—	37	31	—	—	16.2%	—	50	50	—	—	00.0%	—
9	Compound preconditioning	20	20	—	—	00.0%	—	34	32	—	—	05.8%	—	—	—	—	—	—	—
10A	Context sensitivity	33	24	20	17	27.7%	15.0%	1	1	1	1	00.0%	00.0%	—	—	—	—	—	—
10B	Context sensitivity of latent inhibition	50	50	—	—	00.0%	—	1	1	—	—	00.0%	—	—	—	—	—	—	—
11	Generic feedforward multilayer network	—	—	50	50	—	00.0%	—	—	24	21	—	12.5%	—	—	—	—	—	—

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
