# Peer review of "Cortico-Hippocampal Computational Modeling Using Quantum Neural Networks to Simulate Classical Conditioning Paradigms"

_brainsci, 2020, doi:10.3390/brainsci10070431_

Round 1
Reviewer 1 Report
This is a very interesting manuscript that first uses the quantum computation techniques for simulating various aspects of the experimentally-identified neural conditioning paradigms. The authors used the adaptive cortico-hippocampal models with comparison to previously published models, including the author's previous work implemented with conventional neural networks. The present model achieved more reliable results with less trials than the previous ones. The manuscript is well-written and provides details on the methods and results. This reviewer only has few comments for clarity.
Introduction: It's not clearly stated why the authors implement a "lesion" model in the present study, and specific aims and rationale for comparisons with the previous models by Gluck/Myers and Moustafa et al., such as parallel computing.
Methods: Please briefly describe definition of the "steady state" that stops the trial.
Results: Description is rather qualitative and lyrical, and the authors could summarize and emphasize the quantitative aspects of the improvement for speed and efficiency of the present model, as listed in Tables.
There are less improvements of the present model in some tests such as lateral inhibition. Please explain potential reasons on this performance.
Tables 2-4: use of sub-sections (i) and (l) could be avoided to prevent possible confusion, as referring by "intact" and "lesion".
In several places, the authors used the word of "faster" in the computation, but no real data on computation time was presented. Please show quantitative measures, if possible.
There are some typos: 
page 12, Figure 1 ((b)
page 14, Figure 3(f) tasl
Author Response
Response to Reviewer 1 comments
Point 1: This is a very interesting manuscript that first uses the quantum computation techniques for simulating various aspects of the experimentally-identified neural conditioning paradigms. The authors used the adaptive cortico-hippocampal models with comparison to previously published models, including the author's previous work implemented with conventional neural networks. The present model achieved more reliable results with less trials
than the previous ones. The manuscript is well-written and provides details on the methods and results. This reviewer only has few comments for clarity.
We would like to thank the reviewer for his respected efforts to revise our manuscript, appreciate it!
Point 2: Introduction: It's not clearly stated why the authors implement a "lesion" model in the present study, and specific aims and rationale for comparisons with the previous models by Gluck/Myers and Moustafa et al., such as parallel computing.
We revised this part of the introduction by adding the following sentences according to their position in the related paragraph.
• Studying the behavior of both intact and the hippocampal-lesioned systems is significant for the multi-phase learning paradigms. Most of these paradigms have a similar response at the first phase but respond differently to the other phases.
• Comparing our proposed model with the previous models to confirm how CHCQ model performs a wide range of biological paradigms and classical conditioning tasks effectively.
Point 3: Methods: Please briefly describe definition of the "steady state" that stops the trial.
We added the following paragraph to “Proposed model” subsections to clarify the steady state meaning as follows:
However, the output response of the ASLFFQNN is the measured CR which has a value between 0 and 1. During the learning procedure, the value of the output CR varies depending on its related task and condition. Accordingly, this learning ends when the CR value reaches its desired constant state which is the steady state (either 0 or 1).
Point 4: Results: Description is rather qualitative and lyrical, and the authors could summarize and emphasize the quantitative aspects of the improvement for speed and efficiency of the present model, as listed in Tables.
We have updated the following paragraph in Conclusions section for this required revision as
follows:
However, the results presented notable enhancements approved by experimental studies for various tasks, that outperforms the previously published models. A comparison of the CHCQ model with M1 (Gluck/Myers model)[21], M2 (Mustafa et al. model)[22], and G (Green model)[20] models, showed that the CHCQ model has a fast and reliable output CR. Table 2 shows that all the CRs of the CHCQ model reached the final desired states directly after fewer trials than were needed by Gluck/Myers model. Similarly, Table 3 proves that the CHCQ model even takes fewer trial numbers than Moustafa et al. model. Moreover, the CHCQ model completed all of its tasks with more reliable results and more quickly than the Green model as shown in Table 4. It is worth mentioning that the multi–phase learning paradigms of the CHCQ model have no improvement in the first phase as compared to the other models. As long as the first phase of such tasks acts as a pre–exposure phase which takes a similar period in other
models.
Point 5: There are less improvements of the present model in some tests such as lateral inhibition. Please explain potential reasons on this performance.
We have added the following paragraph to the Conclusions section:
It is worth mentioning that the multi-phase learning paradigms of the CHCQ model have no improvement in the first phase as compared to the other models. As long as the first phase of such tasks acts as a preexposure phase which takes a similar period in other models.
Point 6: Tables 2-4: use of sub-sections (i) and (l) could be avoided to prevent possible confusion, as referring by "intact" and "lesion".
Thank you for your notifying us about this point. We added a new row to Tables 2, 3, and 4 indicating the “intact” and “lesioned” related values.
Point 7: In several places, the authors used the word of "faster" in the computation, but no real data on computation time was presented. Please show quantitative measures, if possible.
The word “faster” appeared six times through the manuscript, and we changed all of them to “fewer trial numbers”. And then we readjust the sentence due to the new phrase relatively.
Point 8: There are some typos:
page 12, Figure 1 ((b) à Revised to: Figure 1 (b)
page 14, Figure 3(f) tasl à Revised to: Figure 3 (f) task
Reviewer 2 Report
I feel that the paper correctly criticizes the overuse of artificial neural networks in neuroscience and bring forth quantum computations to address the limitations of artificial neural networks that the proposed quantum computational model attempts to address. There can be little question that quantum computation is required for brain function and the authors experiments to test their model are to be commended.
Author Response
Response to Reviewer 2 comments
Point 1: I feel that the paper correctly criticizes the overuse of artificial neural networks in neuroscience and bring forth quantum computations to address the limitations of artificial neural networks that the proposed quantum computational model attempts to address. There can be little question that quantum computation is required for brain function and the authors experiments to test their model are to be commended.
We would like to thank the reviewer for his respected efforts to revise our manuscript, appreciate it! We have added the following paragraphs to the introduction section for the purpose of this revision:
• Comparing our proposed model with the previous models to confirm how CHCQ model performs a wide range of biological paradigms and classical conditioning tasks effectively.
• However, the usage of quantum computation in the CHCQ model makes it more powerful to simulate the cortico-hippocampal region than the classical computation. Instead of simulating the n-bits input cue of information in classical computation, the quantum computation simulates the same cue as 2n possible states. The quantum circuit
(such as quantum rotation gates) provides computational speedup over the ANNs in classical conditioning simulations. Such accelerated speedup enables the CHCQ model to simulate many biological paradigms efficiently.